# Parameter-Parallel Distributed Variational Quantum Algorithm

Yun-Fei Niu,[1, *] Shuo Zhang,[1, †] Chen Ding,[1] Wan-Su Bao,[1, ‡] and He-Liang Huang[1, 2, 3, 4, §]

[1]*Henan Key Laboratory of Quantum Information and Cryptography, Zhengzhou, Henan 450000, China*
[2]*Hefei National Research Center for Physical Sciences at the Microscale and School of Physical Sciences,*
*University of Science and Technology of China, Hefei 230026, China*
[3]*Shanghai Research Center for Quantum Science and CAS Center for Excellence in Quantum Information and Quantum Physics,*
*University of Science and Technology of China, Shanghai 201315, China*
[4]*Hefei National Laboratory, University of Science and Technology of China, Hefei 230088, China*
(Dated: August 2, 2022)

Variational quantum algorithms (VQAs) have emerged as a promising near-term technique to explore practical quantum advantage on noisy intermediate-scale quantum (NISQ) devices. However, the inefficient parameter training process due to the incompatibility with backpropagation and the cost of a large number of measurements, posing a great challenge to the large-scale development of VQAs. Here, we propose a parameter-parallel distributed variational quantum algorithm (PPD-VQA), to accelerate the training process by parameter-parallel training with multiple quantum processors. To maintain the high performance of PPD-VQA in the realistic noise scenarios, a alternate training strategy is proposed to alleviate the acceleration attenuation caused by noise differences among multiple quantum processors, which is an unavoidable common problem of distributed VQA. Besides, the gradient compression is also employed to overcome the potential communication bottlenecks. The achieved results suggest that the PPD-VQA could provide a practical solution for coordinating multiple quantum processors to handle large-scale real-word applications.

## I. INTRODUCTION

Quantum computing holds the promise of solving certain problems that intractable for classical computers, such as factoring large numbers [1–3], database search [4, 5], solving linear systems of equations [6–8]. However, a universal fault-tolerant quantum computer that can solve efficiently the above problems would require millions of qubits with low error rates [9, 10], which is still a long way from current techniques and may take decades. Thus, we will be in the noisy intermediate-scale quantum (NISQ) era for a long time [11–16]. Variational quantum algorithms (VQAs) leverage a quantum device to minimize a specific cost function [17, 18], by employing a classical optimizer (e.g., Adam optimizer [19]) to train parameter quantum circuits (PQCs). Such algorithms were shown to have natural noise resilience [20] and even benefit from noise, making it particularly suitable for near-term quantum devices, and thus be considered the most promising path to quantum advantage on practical problems in NISQ era [18]. Previous studies have exhibited the application of VQAs on a variety of problems, including classification task [21–24] and generative task [25–27], combinatorial optimization [28–32], quantum many-body problem [33] and quantum chemistry [34–39].

The training process of VQAs is actually not very efficient compared to the classical neural network, due to the following two main reasons: 1) The quantum state of the intermediate process of the quantum circuit cannot be stored, making VQAs impossible to use the backpropagation to update the parameters as efficiently as the classical neural network; 2) A large number of measurements is required for the result read-out of the quantum circuit, which is time-consuming. Therefore, the training of VQAs will face significant challenges, as the amount of data and trainable parameters increases.

To address the above issue, a distributed VQA based on data-parallel has been proposed by Du *et. al.* to accelerate the training of VQA [40]. In this work, a parameter-parallel distributed variational quantum algorithm (PPD-VQA) is proposed to further accelerate the training process by parameter-parallel training with multiple quantum processors. Although the idea of parallel training is not difficult to come up with, including data-parallel or parameter-parallel, it is worth investigating whether the approach works in the realistic scenario that the local quantum nodes will inevitably be affected by quantum noise, and the noise intensity of each node is different. We first proof the convergence of the PPD-VQA, even if each local node has different quantum noise. Further, we design an alternate training strategy to alleviate the acceleration attenuation caused by excessive noise differences among multiple quantum processors, and adopt the gradient compression to cut a large amount of communication bandwidth, to enhance the practicality and scalability of PPD-VQA.

## II. PPD-VQA

The conventional VQAs employ PQCs and update their parameters $\boldsymbol{\theta}$ via a classical optimization training procedure, to find the globle minimum of the given loss functions $L$. Usually, in the training procedure, the gradients of each parameter is evaluated by the parameter-shift rule [41, 42]. The PPD-VQA leverages the fact that the estimates of the gradients of each parameter are genuinely independent of one another at each iteration to accelerate the training of conventional VQA, by parallelizing the gradient estimation across multiple quantum processing unit (QPU) nodes. Conceptually, a classical

* These two authors contributed equally
† These two authors contributed equally; shuoshuo19851115@163.com
‡ bws@qiclab.cn
§ quanhhl@ustc.edu.cn

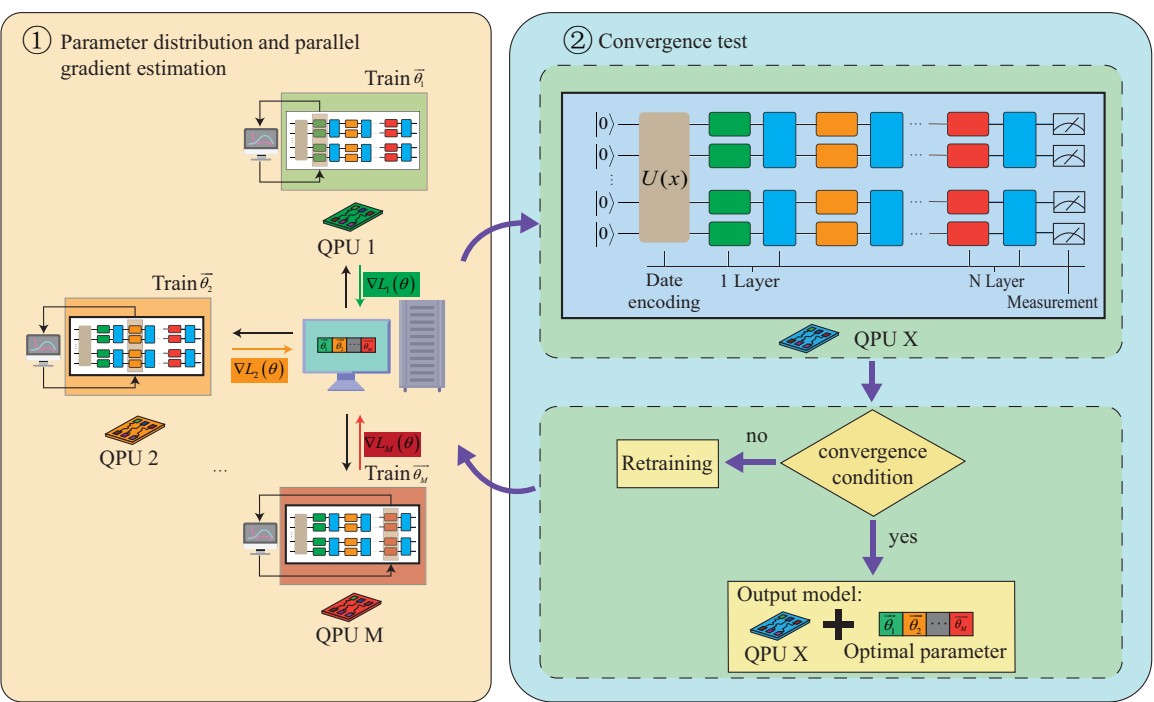

FIG. 1. **Schematic diagram of PPD-VQA.** The diagram illustrates two main steps of the PPD-VQA workflow. Firstly, the central parameter server allocates the trainable parameters to $M$ local nodes, consisting of a QPU and a classic computer, for parallel training. Each local node only trains a part of the trainable parameters, and synchronizes the gradient information to the central parameter server. Secondly, a local node, named QPU X, is selected to verify that the convergence condition is met. If it does not converge, repeat Steps 1 and 2, otherwise, output the optimal parameters and selected local node as the trained model.

central parameter server and $M$ local nodes constitute the framework of PPD-VQA, where each local node consists of a QPU and a classical optimizer. As shown in Fig. 1 and Algorithm 1, at each iteration, the central parameter server divides the trainable parameters $\boldsymbol{\theta}$ into $M$ parts, each of the $M$ local nodes is tasked with computing the gradient of the parameters for a given component. Then, the complete gradient information is obtained through information sharing between local nodes and central parameter sever, which is used to update the trainable parameters as the initial parameters of next iteration. This process is repeated until the optimal parameters are found. The specific process can be divided into the following two steps:

**Step 1: Parameter distribution and parallel gradient estimation.** At the beginning of $t$-th iteration, the classical central server distribute the complete parameter $\boldsymbol{\theta}^{(t)}$ of PQC to each local node as the initial parameters, as well as instructions on which parameters the $i$-th local node is assigned for training. The default instruction is to divide the trainable parameters $\boldsymbol{\theta}^{(t)}$ into $M$ equal parts

$$\boldsymbol{\theta}^{(t)} = [\boldsymbol{\theta}_1^{(t)}, \cdots, \boldsymbol{\theta}_M^{(t)}], \quad \boldsymbol{\theta}_i^{(t)} = \boldsymbol{\theta}^{(t)}[(i-1)\frac{d}{M} : i\frac{d}{M}],$$

and the $i$-th local node is responsible for estimating the corresponding component of gradient $\nabla L_i(\boldsymbol{\theta}^{(t)})$. After the training on each node, the local node synchronizes $\{\nabla L_i(\boldsymbol{\theta}^{(t)})\}_{i=1}^M$ to the central parameter server, and the central parameter server combines the information from each lo-

cal node into a completed gradient $\nabla L(\boldsymbol{\theta}^{(t)})$ used to update $\boldsymbol{\theta}^{(t)}$ to $\boldsymbol{\theta}^{(t+1)}$.

**Step 2: Convergence test.** Choose a local node from the $M$ local nodes, and substitute the parameters $\boldsymbol{\theta}^{(t+1)}$ into this local node. After that, employ this model to the test dataset to to determine whether the convergence condition has been met. The setting of the convergence condition depends on the machine learning task. For example, for the classification task, the convergence condition might be set as a certain classification accuracy threshold. If the convergence condition has not been met, return to Step 1 for the subsequent training iteration; otherwise, output the final parameters and chosen local node as the trained model and terminate the training procedure.

The core idea of PPD-VQA is simple and natural. However, distributed quantum machine learning faces different challenges than its classical counterpart, the main one being that the quantum processors on different local nodes are not identical, due to the inevitable quantum noise. Such non-uniformity manifests itself in two ways: 1) The average noise of each quantum processor is different. For example, some processors have lower noise and some have higher noise; 2) Even if the average noise of each quantum processors is the same, the noise environment of the qubits executing quantum circuits in different processors is unlikely to be consistent. In such a realistic scenario, it remains to be verified whether the parameter-parallel training is still effective, and whether the convergence conditions can be achieved. This important issue is directly related to the practical utility of our scheme and

will be discussed in the next section.

---

**Algorithm 1 The pseudocode of PPD-VQA.**

---

**Require:** $\boldsymbol{\theta} \in [0, 2\pi)^d$: the parameters of ansatz; $L$: loss function; $M$: the number of local nodes, and we donate $M_i$ as the $i$-th local node;

**Ensure:** optimal parameters $\boldsymbol{\theta}^*$

1: **while** convergence condition is not satisfied **do**
2:      The central parameter server divides the parameter $\boldsymbol{\theta}$ into $M$ parts and allocates $\boldsymbol{\theta}$ to $M$ local nodes
3:      **for** Local nodes $M_i, \forall i \in \{1, \cdots, M\}$ in parallel **do**
4:          Calculate gradient component $\nabla L_i(\boldsymbol{\theta})$
5:      **end for**
6:      Synchronize $\nabla L(\boldsymbol{\theta})$ by merging $\{\nabla L_i(\boldsymbol{\theta})\}_{i=1}^M$
7:      Update $\boldsymbol{\theta}$ with a classical optimizer, such as ADAM
8:      Choose a local node from $\{M\}_{i=1}^M$ for convergence test
9:      **if** convergence condition is satisfied **then**
10:          break
11:      **end if**
12: **end while**

---

## III. PERFORMANCE ANALYSIS AND ERROR MITIGATION STRATEGY IN THE REALISTIC NOISE SCENARIO

Gradient represents the optimization direction during the training procedure of VQA, which plays an decisive role in the process of finding the global minima of loss function. Thus, by examining the gradient, we analyze how noise affects convergence of PPD-VQA in the realistic scenario that noise varies for each quantum processor. Furthermore, we will propose a strategy to mitigate the negative consequences that maybe caused by this realistic scenario.

### A. Convergence and acceleration

We apply the "worst-case" noise channel–the depolarizing channels [43] for the following research. According to the Lemma 6 in Ref. [44] all noisy channels $\varepsilon(\cdot)$, which are separately applied to each layer of the ansatz, can be merged together and represented by a new depolarizing channel acting on the whole ansatz, i.e.,

$$\tilde{\varepsilon}(\rho) = (1 - \tilde{p})\rho + \tilde{p}\frac{\mathbb{I}}{2^n} \tag{1}$$

where $\tilde{p} = 1 - (1 - p)^N$, $p$ is the depolarizing probability in $\varepsilon(\cdot)$, and $N$ refers to depth of ansatz. Obviously, the depolarizing noise turns the quantum state into a maximally mixed state with a certain probability, which could make the gradients obtained by parameter-shift-rule in the experiment deviate from that of the ideal environment without noise.

Now we quantify the convergence of PPD-VQA with multiple local nodes that have different performance, by using the following utility metric [44]:

$$R_1(\boldsymbol{\theta}^{(T)}) = \mathbb{E}[\|\nabla L(\boldsymbol{\theta}^{(T)})\|^2] \tag{2}$$

where $T$ is the number of iterations and the expectation $\mathbb{E}[\bullet]$ is taken over the randomness of depolarizing noise and measurement error. This metric evaluates how far the result is away from the stationary point. The upper bounds of $R_1(\boldsymbol{\theta}^{(T)})$ when implementing PPD-VQA with multiple non-identical processors are summarized in the following theorem.

**Theorem 1** *Suppose that we employ the mean square error (MSE) loss function,* $L = \frac{1}{2N_D}\sum_k (\hat{y}_k - y_k)^2 + \frac{\lambda}{2}\|\boldsymbol{\theta}^{(t)}\|_2^2$, *where* $\hat{y}_k = \langle O \rangle$ *is the predicted label with* $\langle O \rangle$ *being the outcome of the observable $O$ by $K$ measurements, $N_D$ is the number of the data, and $\lambda \geq 0$ is the regularizer coefficient, $M$ noisy local nodes of PPD-VQA have different depolarizing noise with depolarizing probability $\{\tilde{p}_i\}_{i=1}^M$, the metric $R_1(\boldsymbol{\theta}^{(T)})$ has following upper bound*

$$\begin{aligned} R_1 \leq &\frac{1 + 9\pi^2\lambda d}{2T(1 - \tilde{p}_{max})^2} \\ &+ \frac{2G + d}{(1 - \tilde{p}_{max})^2}(2 - \tilde{p}_{max})\tilde{p}_{max}(1 + 10\lambda)^2 \\ &+ \frac{2dK + d}{2N_D K^2}\frac{1}{(1 - \tilde{p}_{max})^2}. \end{aligned}$$

*where loss function $L$ is $S$-smooth with $S = (3/2 + \lambda)d^2$, $G$-Lipschitz with $G = d(1 + 3\pi\lambda)$, and $\tilde{p}_{max} = \max\{\tilde{p}_i\}_{i=1}^M$.*

The proof of Theorem 1 is essentially similar with conventional VQA, for both of them acquire the complete gradient information only once in one iteration. Therefore, one can obtain the upper bound of $R_1(\boldsymbol{\theta}^{(T)})$ of PPD-VQA in noise scenario by straightforward following the proof procedure of Theorem 1 in Ref. [44]. We briefly sketch our proof as follows.

The first step is to establish the relation between the $j-$th component of the analytical gradients $\nabla_j L(\boldsymbol{\theta}^{(t)})$ (unbiased) and the estimated gradients $\nabla_j \bar{L}_i(\boldsymbol{\theta}^{(t)})$ (biased) that is evaluated from QPU $i$, which can be directly obtained from Ref. [44]

$$\nabla_j \bar{L}_i(\boldsymbol{\theta}^{(t)}) = (1 - p_i)^2 \nabla_j L(\boldsymbol{\theta}^{(t)}) + C_{i,j}^{(t)} + \varsigma_{i,j}^{(t)} \tag{3}$$

where the biased term $C_{i,j}^{(t)}$ originates from the depolarizing noise, and the zero mean random variable $\varsigma_{i,j}^{(t)}$ is from both the depolarizing noise and measurement error. Then one can further utilize the $S$-smooth and $G$-Lipschitz of the $L$ to calculate the loss difference, i.e.,

$$\begin{aligned} &L(\boldsymbol{\theta}^{(t+1)}) - L(\boldsymbol{\theta}^{(t)}) \\ \leq &\langle \nabla L(\boldsymbol{\theta}^{(t)}), \boldsymbol{\theta}^{(t+1)} - \boldsymbol{\theta}^{(t)} \rangle + \frac{S}{2}\|\boldsymbol{\theta}^{(t+1)} - \boldsymbol{\theta}^{(t)}\|_2^2 \end{aligned} \tag{4}$$

Substitute Eq.(3) and $\boldsymbol{\theta}^{(t+1)} = \boldsymbol{\theta}^{(t)} - \eta\sum_{ij}\nabla_j\bar{L}_i(\boldsymbol{\theta}^{(t)})$ (we set the learning rate $\eta = 1/S$) into Eq.(4) and take the expec-

tation over the random variable $\varsigma_{i,j}^{(t)}$, one have

$$
\begin{aligned}
&\mathbb{E}_{\varsigma_{i,j}^{(t)}}[L(\boldsymbol{\theta^{(t+1)}}) - L(\boldsymbol{\theta^{(t)}})] \\
&\leq \sum_{i,j} \left[ -\frac{1}{2S}(1-\tilde{p}_i)^2 \left(\nabla_j L(\boldsymbol{\theta^{(t)}})\right)^2 \right. \\
&\left. \quad + \frac{2G/d+1}{2S}(2-\tilde{p}_i)\tilde{p}_i(1+10\lambda)^2 \right] \\
&\quad + \frac{2dK+d}{4SN_D K^2}
\end{aligned}
\tag{5}
$$

Note that $-\sum_{i,j}(1 - \tilde{p}_i)^2 \left(\nabla_j L(\boldsymbol{\theta^{(t)}})\right)^2 \leq -(1 - \tilde{p}_{max})^2 \|\nabla L(\boldsymbol{\theta^{(t)}})\|^2$, and $(2-\tilde{p}_i)\tilde{p}_i \leq (2-\tilde{p}_{max})\tilde{p}_{max}$, we obtain

$$
\begin{aligned}
&\|\nabla L(\boldsymbol{\theta^{(t)}})\|^2 \\
&\leq 2S \frac{L(\boldsymbol{\theta^{(t)}}) - \mathbb{E}_{\varsigma_{i,j}^{(t)}} L(\boldsymbol{\theta^{(t+1)}})}{(1-\tilde{p}_{max})^2} \\
&\quad + \frac{2G+d}{(1-\tilde{p}_{max})^2}(2-\tilde{p}_{max})\tilde{p}_{max}(1+10\lambda)^2 \\
&\quad + \frac{2dK+d}{4SN_D K^2}\frac{1}{(1-\tilde{p}_{max})^2}
\end{aligned}
\tag{6}
$$

Finally, by summing over $t = 0, 1, \cdots, T$, the upper bound of $R_1(\boldsymbol{\theta^{(T)}})$ is achieved.

From Theorem 1 above and Theorem 1 in Ref. [44], we can observe that the convergence rate between conventional VQA and PPD-VQA is similar, i.e., both of them scale with $O(1/\sqrt{T})$ [44], since the second term and the third term are constant in above inequality when $\{\tilde{p}\}_{i=1}^M$ is fixed. The similar convergence rate guarantees that PPD-VQA promises a intuitive linear runtime speedup with respect to the increased number of local nodes $M$.

Next, we perform numerical experiment to study the performance of PPD-VQA in the realistic noise scenario. In our simulations, we apply PPD-VQA to the binary classification task, by employing the Iris dataset and ansatz shown in Fig. 2. We choose 100 examples from Iris dataset with 50 versicolors (label 0) and 50 virgunicas (label 1), where 75% examples are randomly selected as the training set and the remaining 25% as the test set. We implement the task using the PPD-VQA with $M =$1 (conventional VQA), 2, 4, 8 local nodes, respectively. For each type of PPD-VQA, we also set different noise parameters separately. Specifically, for each node the PPD-VQA, the depolarizing probability $p_i$ for single-qubit gate is set by sampling from a Gaussian distribution i.e., $p_i \sim N(\mu, \sigma^2)$, where the mean $\mu$ varies from 0.01 to 0.05 with step 0.02 and $\sigma = \mu/9$. The depolarizing probability of two-qubit gate is set as $4p_i$ refer to the performance of SOTA quantum processor $Zuchongzhi$ [15]. Each local node's noise will be somewhat different as a result of such random sampling. A total of 100 independent experiments were run for each setting, and in each experiment, the measurement shots is set to 8192, batchsize is set to 5, and the convergence condition is that the classification accuracy

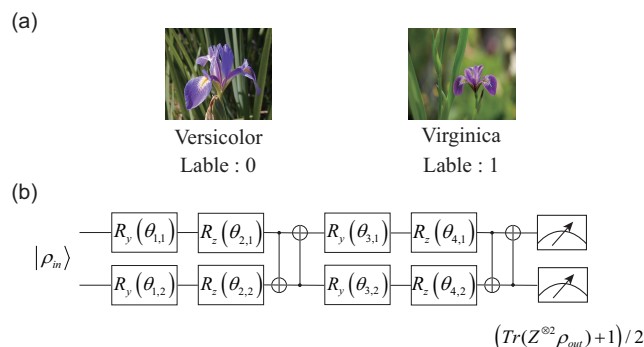

(a)

Versicolor
Lable : 0

Virginica
Lable : 1

(b)

$|\rho_{in}\rangle$ — $R_y(\theta_{1,1})$ — $R_z(\theta_{2,1})$ — $R_y(\theta_{3,1})$ — $R_z(\theta_{4,1})$ —

— $R_y(\theta_{1,2})$ — $R_z(\theta_{2,2})$ — $R_y(\theta_{3,2})$ — $R_z(\theta_{4,2})$ —

$\left(Tr(Z^{\otimes 2}\rho_{out})+1\right)/2$

FIG. 2. **Classification task on Iris dataset and the ansatz in numerical simulation.** (a) A visualization of training examples sampled from Iris dataset. We choose the data of iris versicolor (lable 0) and iris virginica (lable 1) for binary classification. (b) Ansatz of PPD-VQA for the classification. We encode the classical example $x_k$ in Iris dataset, which has 4 attributes, into the quantum state $\rho_k$ by amplitude encoding. Then a hardware-efficient PQC with trainable single-qubit gate is employed for the training. The quantum observable is set as $Z^{\otimes 2}$, and the measurement result is mapped to $[0, 1]$.

on the training set exceeds 96%. As shown in Fig. 3(a, b), for both conventional VQA and PPD-VQAs with 2, 4, and 8 local nodes, the number of iterations required to achieve a preset training accuracy increases with the mean of noise $\mu$, and the PPD-VQA with multiple local nodes has a similar convergence speed as conventional VQA (see Fig. 3(b)), which is consistent with Theorem 1.

We further introduce a metric, i.e. $R_S = T_1/T_m$ to evaluate the speed-up ratio of the PPD-VQA with $M = m > 1$ local nodes compared to the conventional VQA with just $M = 1$ local node, where $T_1$ and $T_m$ are the time consuming of conventional VQA and PPD-VQA from the start of training to meeting the convergence conditions, respectively. Assuming that the time consumption of implementing each quantum circuit is the same (since the number of measurements is the same, and only the rotation angle of the single-qubit gate will be changed each time the circuit is executed), the formula of the speedup ratio $R_S$ can be further rewritten as[1]:

$$
R_S = \frac{(1+2d) \times N_D \times N_I^1}{(1+\frac{2d}{M}) \times N_D \times N_I^M} = \frac{(1+2d) \times N_I^1}{(1+\frac{2d}{M}) \times N_I^M}, \tag{7}
$$

where $d$ is the number of parameters, $N_E^1$ and $N_E^M$ are the total number of iterations for the conventional VQA and PPD-VQA, respectively. In the ideal scenario of noiseless,

---

[1] According to parameter-shift-rule, $\nabla_j \bar{L}_i(\boldsymbol{\theta^{(t)}})$ ( $j$-th component of parameters vector $\boldsymbol{\theta^{(t)}}$) satisfies $\frac{1}{N_D}\left[\sum_k (\hat{y}_k^{(t)} - y_k)\frac{\hat{y}_k^{(t,+j)} - \hat{y}_k^{(t,-j)}}{2} + \lambda\theta_{i,j}^{(t)}\right]$, where $\hat{y}_k^{(t,\pm j)}$ donates the output of PPD-VQA with shifted parameter $\boldsymbol{\theta^{(t)}} \pm \frac{\pi}{2}\boldsymbol{e_{i,j}}$, and $\boldsymbol{e_{i,j}}$ donates the unit vector. Thus, for each data, the local node should implement $1 + 2d/M$ quantum circuits for the gradient estimation, where $d/M$ is the number of parameter in each local node.

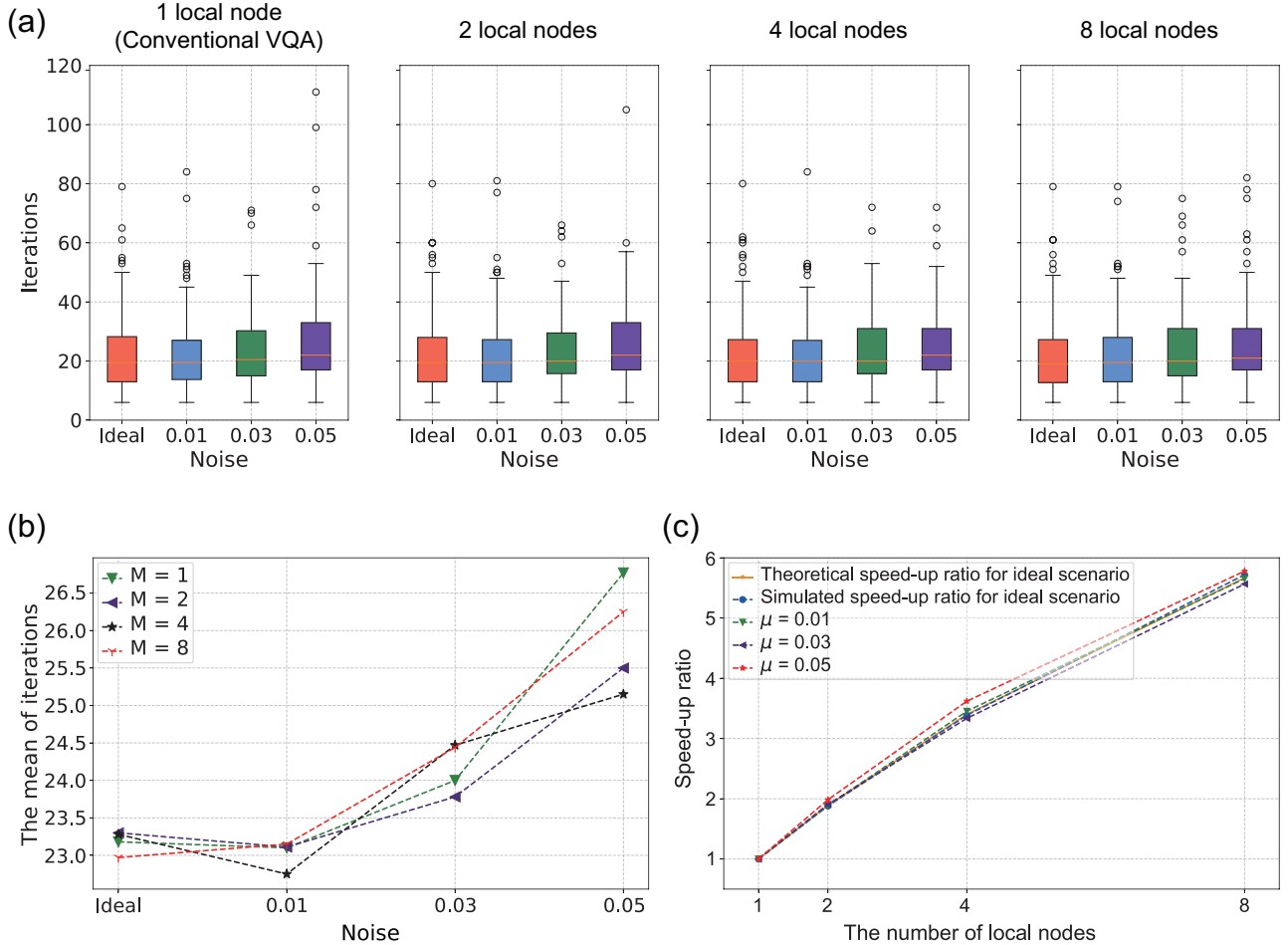

FIG. 3. **Simulation results of PPD-VQA with $M$ local nodes under noise scenario for Iris dataset classification.** (a) Boxplots count the iterations of PPD-VQA with $M$ local nodes, where $M = 1, 2, 4, 8$ from left to right, when achieving a predefined training accuracy. The depolarizing probability $p_i$ for single-qubit gate is set by sampling from a Gaussian distribution i.e., $p_i \sim N(\mu, \sigma^2)$, where the mean $\mu = 0$ (ideal case), $0.01, 0.03, 0.05$, and $\sigma = \mu/9$. (b) Scaling behavior of the mean of the iterations in (a) for increasing noise ($\mu$). The results of PPD-VQA with $M = 1, 2, 4, 8$ local nodes are shown. (c) Scaling behavior of speed-up ratio in clock-time for increasing number of local nodes $M$. The results of different depolarizing probabilities are shown.

$N_E^1 = N_E^M$, thus $R_S = \frac{1+2d}{1+\frac{2d}{M}}$ in ideal scenario. Figure 3(c) shows that speed-up ratio for the PPD-VQA with 1 (conventional VQA), 2, 4, 8 local nodes under under a variety of noise scenarios. No matter how the mean of noise $\mu$ changes, the speed-up ratio of PPD-VQA is almost only related to the number of local nodes and is extremely close to the ideal case. This result strongly supports that PPD-VQA can achieve a very good acceleration in realistic scenarios.

**B. Alternate training strategy for mitigating the negative effects of large noise differences between different local nodes**

In the previous subsection, the difference in the noise of the quantum processors of each node is not particularly large, because the noise is set by sampling from a Gaussian distribution $N(\mu, \sigma^2)$, where $\sigma = \mu/9$. In this subsection, we will study the performance of PPD-VQA in cases where the noise

difference is more pronounced.

We first monitor the performance of PPD-VQA when the noise difference of different local nodes changes from small to large. To quantify the noise differences of local nodes, we introduce a metric, named $Differ$, which is defined as

$$Differ = D_{KL}(P(p) \| P_{\text{Uniform}}),$$

where $D_{KL}$ is Kullback-Leibler (K-L) divergence [45], $P_{\text{Uniform}}$ refers to the uniform distribution, $P(p)$ is the normalized distribution of depolarization probability of each local node, where $P(p)_k = p_k / \sum_{i=1}^{M} p_i$, and $p_k$ is the depolarization probability of the $k$-th local node. With this metric, a noise setting with a resulting distribution that corresponds to a higher K-L divergence with respect to uniform distribution would mean greater noise variance between local nodes. Besides, we set another constraint that the mean of $\{p_i\}_{i=1}^{M}$ is 0.04. For each PPD-VQA with $M \in [1, 2, 4, 8]$ local nodes, $Differ$ varies from 0 to 0.625, we generate 10 instances of

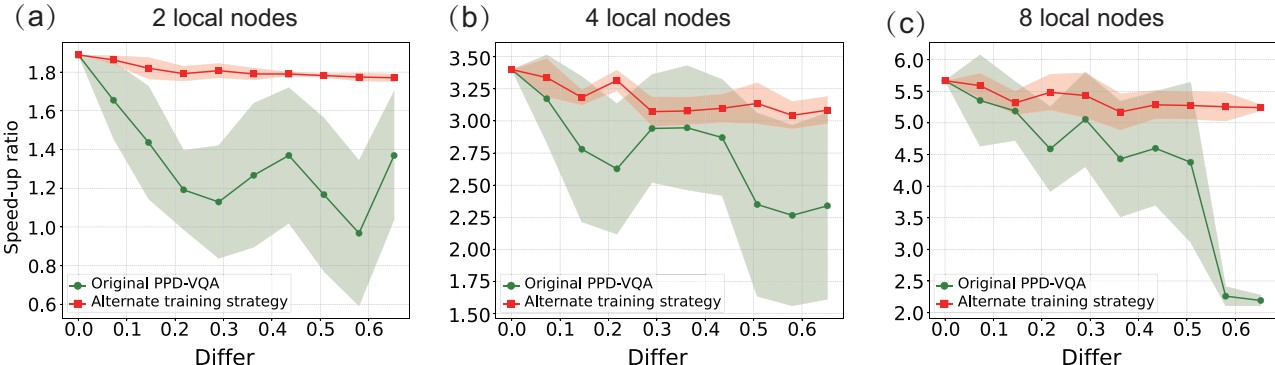

FIG. 4. **Simulation results of PPD-VQA with $M$ local nodes in cases where the noise difference is more pronounced.** (a) The average speed-up ratio as a function of $Differ$ (a metric for quantifying the noise differences of local nodes) for the PPD-VQA with $M = 2$(a), 4(b), 8(c) local nodes. 100 independent experiments are implemented for each setting. The green and red solid lines present the results for original PPD-VQA and PPD-VQA with alternate training strategy, respectively. It is obvious that the red curves have noticeable larger values and smaller variances than the green curves in all three cases.

noise setting for each $Differ$, and for each instance 50 experiments with different initial parameters are implemented. As shown in Fig. 4, the speed-up ratio tends to become smaller as the $Differ$ increases, indicating that the advantage of PPD-VQA in terms of speedup is diminished in extreme cases where the noise difference between local nodes is significant.

To suppress acceleration decay of PPD-VQA caused by excessive noise difference between local nodes, we propose a simple but effective approach named as alternate training strategy, whose core idea is decoupling the trainable parameter groups and corresponding quantum processors. The process of this alternate training strategy is as follows: Suppose that at the first iteration, the $i$-th local node is scheduled to train the parameters $\theta_i$. We denote this process as $\{\theta_i : \mathrm{QPU}_i\}_{i=1}^M$. Then in the next iteration, The corresponding relationship between trainable parameters and local nodes becomes $\{\theta_M : \mathrm{QPU}_1\} \cup \{\theta_i : \mathrm{QPU}_{i+1}\}_{i=1}^{M-1}$, that is, we perform a cyclic shift on the correspondence between the trainable parameters and local nodes. The alternate training strategy is repeated with the training process, which makes each parameter group $\theta_i$ be trained in turn by all quantum processors throughout the whole training process. The numerical simulation results of PPD-VQA with alternate training strategy is shown in Fig. 4. An immediate observation is that when the noise difference between local nodes increases, PPD-VQA performance degrades relatively little thanks to the alternate training strategy. Besides, the performance of PPD-VQA becomes more stable as the variance of the mean of different experiments is significantly smaller. These two benefits suggest that this strategy can be effectively employed for mitigating the negative effects of large noise differences between different local nodes.

## IV. GRADIENT COMPRESSION

Another challenge of distributed machine learning is the large amount of communication bandwidth for gradient exchange [46]. With the development of quantum computing

hardwares, this problem may also arise in large-scale distributed quantum machine learning. To overcome this potential problem, we adopt the technique of gradient compression [47] widely used in the classical community to PPD-VQA, to reduce the communication bandwidth for distributed training. The pseudocode of PPD-VQA with gradient compression for local node $i$ in PPD-VQA is as follows.

---

**Algorithm 2 The pseudocode of PPD-VQA with gradient compression.**

---

**Require:** $\theta \in [0, 2\pi)^d$: the parameters of ansatz; $L$: loss function; $M$: the number of local nodes, and we donate $M_i$ as the i-th local node; $Mask = (0, \cdots, 0)$ has the same dimension with $\theta_i$ defined in section *PPD-VQA*, and $\odot$ is hadamada product, i.e., $\boldsymbol{a} \odot \boldsymbol{b} = (a_1 b_1, \cdots, a_n b_n)$
**Ensure:** optimal parameters $\theta^*$
1: Calibrate thershold $thr$
2: $\nabla L_i(\boldsymbol{\theta}) = \mathbf{0}$ for $i \in [1, 2, \cdots, M]$
3: **while** convergence condition is not satisfied **do**
4:     The central parameter server divides the parameter $\boldsymbol{\theta}$ into $M$ parts and allocates $\boldsymbol{\theta}$ to $M$ local nodes
5:     **for** Local nodes $M_i, \forall i \in [M]$ in parallel **do**
6:         Calculate gradient component $G_i(\boldsymbol{\theta})$
7:         $\nabla L_i(\boldsymbol{\theta}) = \nabla L_i(\boldsymbol{\theta}) + G_i(\boldsymbol{\theta})$
8:         **for** $j = (i-1)\frac{d}{M}, \cdots, i\frac{d}{M}$ **do**
9:             **if** $|\nabla_j L_i(\boldsymbol{\theta})| > thr$ **then**
10:                 $Mask[j] = 1$
11:             **end if**
12:         **end for**
13:         $g_i(\boldsymbol{\theta}) = \nabla L_i(\boldsymbol{\theta}) \odot Mask$
14:         $\nabla L_i(\boldsymbol{\theta}) = \nabla L_i(\boldsymbol{\theta}) \odot \neg Mask$
15:     **end for**
16:     Synchronize $g_i(\boldsymbol{\theta})$ by merging $\{g_i(\boldsymbol{\theta})\}_{i=1}^M$;
17:     Update $\boldsymbol{\theta}$ with a classical optimizer, such as ADAM;
18:     Choose a local node from $\{M\}_{i=1}^M$ for convergence test.
19:     **if** convergence condition is satisfied **then**
20:         break
21:     **end if**
22: **end while**

---

The idea of gradient compression is gradient clipping,

| M | noise setting | without gradient compression | | with gradient compression | | result | |
|---|---|---|---|---|---|---|---|
| | | iterations | communication volume | iterations | communication volume | compression ratio | speed-up ratio |
| 2 | $\mu = 0.016$ | 2324 | 2324×8 | 2259 | 7016 | 62.3% | 1.87 → 1.92 |
| | $\mu = 0.064$ | 2680 | 2680×8 | 2780 | 8565 | 60.1% | 2.04 → 1.97 |
| 4 | $\mu = 0.016$ | 2324 | 2324×8 | 2444 | 3630 | 80.0% | 3.36 → 3.20 |
| | $\mu = 0.064$ | 2712 | 2712×8 | 3295 | 2862 | 86.9% | 3.64 → 2.99 |
| 8 | $\mu = 0.016$ | 2282 | 2282×8 | 2463 | 3634 | 80.0% | 5.71 → 5.29 |
| | $\mu = 0.064$ | 2702 | 2702×8 | 3200 | 2818 | 87% | 6.09 → 5.14 |
| 2 | $\mu = 0.016$ | 2324 | 2324×8 | 5659 | 701 | 96.3% | 1.87 → 0.77 |
| | $\mu = 0.064$ | 2680 | 2680×8 | 6463 | 787 | 96.3% | 2.04 → 0.84 |
| 4 | $\mu = 0.016$ | 2324 | 2324×8 | 6338 | 623 | 96.7% | 3.36 → 1.23 |
| | $\mu = 0.064$ | 2712 | 2712×8 | 6410 | 791 | 96.4% | 3.64 → 1.54 |
| 8 | $\mu = 0.016$ | 2282 | 2282×8 | 6043 | 607 | 96.7% | 5.71 → 2.15 |
| | $\mu = 0.064$ | 2702 | 2702×8 | 6322 | 781 | 96.4% | 6.09 → 2.60 |

TABLE I. **The comparison of the performance between PPD-VQA without gradient compression and with gradient compression.** The results of PPD-VQA under different number of local nodes ($M = 2$, $M = 4$ and $M = 8$) and noise settings ($\mu = 0.016$ and $\mu = 0.064$) are presented. In the table we count total number of iterations for all 100 instances in each setting. Communication volume $CV$ is defined as the total number of gradient components after clipping transmitting between the central parameter server and multiple local nodes, and compression ratio is $1 - CV_{\text{with}}/CV_{\text{without}}$, where $CV_{\text{with}}$ ($CV_{\text{without}}$) is the communication volume for PPD-VQA with (without) gradient compression. The symbol $\rightarrow$ indicates the change of speed-up ratio from left (PPD-VQA without gradient compression) to right (PPD-VQA with gradient compression). Two typical results help us explore the relationship between acceleration of PPD-VQA and compression ratio: (top) The acceleration of PPD-VQA with gradient compression has only a slight decay when the gradient compression ratio is over 60%. (bottom) The acceleration of PPD-VQA with gradient compression decreases significantly when the gradient compression ratio is too high (over 96%).

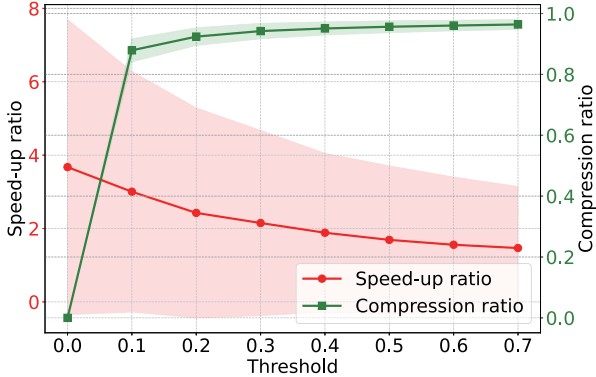

FIG. 5. **The speed-up ratio and compression ratio as a function of threshold value for the PPD-VQA with $M = 4$ local nodes.** The depolarizing noise $p_i$ in each node is set by sampling from the gaussian distribution $N(\mu, \sigma^2)$ with $\mu = 0.016$ and $\sigma = \mu/9$.

which makes the gradient sparse by comparing its individual components with a threshold $thr$. Only the components of the gradient with larger absolute values compared with $thr$ can be synchronized to the central parameter server, thus ensuring that the general direction of the parameter update remains correct. The remaining components smaller than $thr$ are still retained in corresponding local node and counted as a part of new gradient in next iteration. Thus we obtain the uncropped original $\nabla L_i(\boldsymbol{\theta})$ in local node $i$. This method greatly reduces

the actual communication bandwidth required in PPD-VQA. However, due to the existence of quantum noise, it is also unknown whether gradient compression works on PPD-VQA, so next we will perform numerical simulations to address this concern.

We test the gradient compression on a PPD-VQA with $M = 4$ local nodes, where the noise $p_i$ in each node is set by sampling from the gaussian distribution $N(\mu, \sigma^2)$ with $\mu = 0.016$ and $\sigma = \mu/9$. In our simulation, the threshold value $thr$ varies from 0 to 0.7 with step 0.1, and we still implement 100 independent experiments for each setting. As shown in Fig. 5, by setting a reasonable compression threshold, we can greatly reduce the communication cost. It can be also observed that the increase of the gradient compression ratio leads to the decay of the acceleration of PPD-VQA. When $thr > 0.1$, the growth of gradient compression ratio becomes very slow, while speed-up ratio is still decreasing rapidly. Thus, we need to find a balance between the decay of acceleration advantage and reducing the communication volume. When the threshold value is 0.1, we can achieve a relatively high gradient compression ratio ($> 80\%$) without losing too much acceleration advantage ($R_S > 2.7$).

In Table I, we further show two types of typical results for the PPD-VQA with $M = 2, 4, 8$ local nodes, and noise level $\mu = 0.016, 0.064$. For each setting, 100 independent experiments are implemented. In the first typical result, we set a reasonable compression ratio, so that the speed-up ratio is almost not lost compared to the uncompressed case. However, in this scenario, the compression ratio can still be higher than 60%, or even up to 87%, indicating that we can solve the problem on

communication bottleneck without losing too much acceleration advantage of PPD-VQA. In the second typical result, to achieve a more aggressive compression ratio (above 96%), the speedup of PPD-VQA is significantly reduced. Possible reason is that fewer trainable parameters make the gradient not have the sparsity compared with deep neural networks, which leads to a significant increase in the number of iteration when we apply the gradient compression algorithm to PPD-VQA. Anyway, our experiments demonstrate that gradient compression is very suitable for PPD-VQA, even in the realistic noise scenario.

In addition to reducing communication bandwidth, the gradient compression has another benefit for PPD-VQA, that is, reducing the error caused by quantum operations. Since small gradients are forbidden to update, few parameters in the quantum circuit need to be changed in each iteration. For some experimental systems, such as free-space linear optical quantum platforms, each time the parameters are changed, some optical components need to be rotated, so reducing the number of rotations can reduce the accumulation of operation errors.

## V. CONCLUSION

Our results show that PPD-VQA is highly promising as it achieves approximately linear acceleration over the training process of conventional VQA, both in theory and simulation results. The PPD-VQA exhibits good resilience to the excessive noise differences among local nodes, by employing the alternate training strategy. Furthermore, by adopting the gradient compression strategy, potential communication bottlenecks can also be addressed to support the future scalability of PPD-VQA.

The PPD-VQA is naturally compatible with data-parallel distributed VQA proposed in [40], so the combination of the two approaches could enable a stronger acceleration for the training of VQA. Besides, PPD-VQA may face the same dilemma as conventional VQAs, such as barren plateaus [48–54], generalization ability and trainability [55], which requires more in-depth discussions in the future works. Some more complex application scenarios, such as privacy-preserving distributed VQA, are also worth studying.

## ACKNOWLEDGMENTS

H.-L. H. acknowledges support from the Youth Talent Lifting Project (Grant No. 2020-JCJQ-QT-030), National Natural Science Foundation of China (Grants No. 11905294), China Postdoctoral Science Foundation, and the Open Research Fund from State Key Laboratory of High Performance Computing of China (Grant No. 201901-01).

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
