# Peer review of "Parameter-Parallel Distributed Variational Quantum Algorithm"

_SciPost Physics_

## Round 1 · Referee Report · Anonymous · 2022-9-22

Report

The author bring forward an algorithm for variational quantum computing in which they parallelize the evaluation of gradients of the parameters over different quantum processing units (QPUs). The authors fairly and clearly state that the idea in itself is not groundbreaking, however, what makes this work interesting and useful is that they evaluate the performance of such parallelziation in the presence of noise. It could be in fact possible that the idea would not work well when each QPU is noisy, and in its own way.
The authors show that even in presence of noise there is a significant speed up from parallelization, and actually the noisy systems scale as well as the ideal one.
For the case of QPUs with significantly different noise, the authors showed also that one can rotate through the QPUs used for different set of parameters, doing so "averaging out" the different noise scenarios for each set of parameters and thus performing in a more consistent way.
They also evaluate the performance of gradient clipping, which they use to remove gradients which are too small and thus reducing the communication bottleneck. In this case they show that even reductions of 60% result in almost no change in the speed-up.

Some comments:

1) When introducing the issues faced by VQA, the authors do not mention barren plateaux, which are then mentioned in the conclusions. Is there a particular reason for this choice?

2) Regarding the alternate training strategy, it seems to me that the more the setup is parallelized the closer becomes to the first training proposed. If this is true, it could imply that if one parallizes more, he/she should not need the alternate training. What do the authors think about this?

3) Do the authors have an idea of why the speed-up seems to be unaffected by the presence of noise? The curves in Fig.3(b,c) are remarkably close to each other.

4) It is not clear to me how to compare the values of the noise used with those found on QPUs like Zuchongzhi. Can the authors say a bit more about this so that we can understand better when such study would be really applicable to available QPUs?

5) Regarding the classification task, I think that more information could be provided to the broad audience who could read this article. For instance how are the data encoded classically? How is it converted to be used on the quantum machine? More clarity could be given on the number of parameter used. There is some description in the caption, but I think that it could be expanded in the main text.

6) Fig.1. For M-th QPU, I think that the portion that is optimized should be color-coded in red, but it is in orange.

7) Fig.3(a). It is not clear to me what the various error bars, colored areas and circles mean. More detail and explanation could help.

8) Fig.4. It is not clear to me what the shaded area exactly represents

9) The english would need a bit of polishing, as there are a lot of minor issues, but this is something that can be done at a later stage.

To conclude, I do not think that the paper is ready to be published in Scipost, and I think that the authors may have to present a more convincing case for the paper to be published in Scipost Physics instead of Scipost Physics Core. In fact I am not convinced that the paper satisfy at least one of the 4 criteria
1. Detail a groundbreaking theoretical/experimental/computational discovery;
2. Present a breakthrough on a previously-identified and long-standing research stumbling block;
3. Open a new pathway in an existing or a new research direction, with clear potential for multipronged follow-up work;
4. Provide a novel and synergetic link between different research areas.

  • validity: good
  • significance: good
  • originality: good
  • clarity: good
  • formatting: good
  • grammar: below threshold

---

## Round 1 · Referee Report · Hao Wang · 2022-10-31

Strengths

1. The general concept of distributed gradient computation is good.

2. Also, a simple yet working error mitigation method is devised to average out the bias in estimated sub-gradient across local QPUs.

Weaknesses

1. The theoretical analysis seems not to consider the bias in estimating the sub-gradient locally on QPUs.

2. The theoretical analysis is not new, which is entirely based on previous works.

3. The discussion on how differences in noise models across QPUs affect the convergence speed is a bit shallow.

Report

This manuscript proposes distributing the computation of sub-gradients of the loss function of VQAs to multiple QPUs to speed up the wall clock time. As a straightforward approach applied extensively in classical machine learning, this idea has not been tested in QML.
The authors show the anticipated wall-clock time speedup with simulated gate noises on a straightforward classification problem using a shallow hardware-efficient ansatz. Also, the authors prove the convergence speed of the gradient-based optimizer in the distributed setting has the same upper bound as in the traditional VQAs.
Moreover, it is empirically illustrated that when the noise distributions differ significantly across local QPUs, the gradient-based optimizer's convergence speed is hampered quite a bit, for which the authors propose a rotation rule for averaging out the discrepancy of the noises across different QPUs.

I have the following major concerns:

1. I think the theoretical part is not very satisfactory, primarily based on Ref. [44], as you pointed out in the paper. It is crucial to look at the structure of the noise closely, i.e., the covariance of the estimated partial derivatives, i.e., $Cov\{\partial \hat{L}/\partial \theta_i, \partial \hat{L}/\partial \theta_j\}$ ("hat" means it is estimated with parameter-shift rule), due to quantum noises. In the traditional, centralized setup, this covariance is often used to mitigate errors (at least measurement errors) for the gradient-based methods. In the distributed setup, I presume that the covariance admits a block-diagonal structure (since local QPUs are independent), which should play a role in your analysis.

2. It is unclear what you meant by "convergence test." Is it nothing more than selecting a QPU u.a.r. and getting the accuracy score on the classification task?

3. IMHO, the gradient compression part is superfluous, which is indeed a bottleneck in deep learning, where we have to face millions of parameters. However, in VQAs, we cannot afford that, correct? due to the current limitation of the hardware implementation and also to the theory that says barren plateaus will kick us out of the game if we have $O(poly(n))$ layers (n is the number of qubits). Therefore, I do not think gradient compression is needed. (Yes, in table I, you show some compression results; but I think you are saving the communication cost from a small overhead)

4. The most interesting aspect to me is Eq. (3), which writes down the biased of the estimated gradient on each local node/QPU. I think the author should investigate the relationship between the noise level and the bias term, which can drastically change the gradient direction if the noise is high and the magnitude of the sub-gradient is small.

5. Also, in light of the above question, I wonder if the distributed scheme converges at all. Does the bias term also scale down with the diminishing sub-gradient/partial derivative when approaching the critical points on the quantum loss landscape? Otherwise, if the partial derivatives go to zero while the bias does not, we could have a serious problem. Maybe I overthink this part, as perhaps the bias is so tiny that it can be ignored. Please comment on this.

6. Do you contemplate any error mitigation approach on each local node?

7. Please improve some usage of the language/jargon.

* "..the estimates of the gradients of each parameter.." -> the gradients/partial derivatives of the observable w.r.t. to each parameter

* I don't like the notation used in the expression under "Step 1". It is more standard in math to express it by $\boldsymbol{\theta}^{(t)}_i = (\theta^{(t)}_{1 + (i-1)n}, \theta^{(t)}_{2 + (i-1)n}, \ldots, \theta^{(t)}_{i n}), n = d / M.$

* Also, please define the loss function/expectation L first (should simply take one sentence).

* $\nabla L_i(\boldsymbol{\theta}^{(t)})$ is confusing/non-standard to me, which immediately implies you have a sub-function L_i of L. I assume this means the loss function is defined on a batch of data sets, correct? Please either provide an explicit definition thereof or use a more understandable notation. Provided that I understand it correctly, the gradient step should also be divided by the batch size, right? See equation $\theta^{(t+1)} = \theta^{(t)} - \eta \sum_{ij}\nabla_j$ ... at the very bottom of page 3.

* Please clarify if the random variables \textvarsigma on the same QPU are independent, which is essential for the upper bound derived in the paper.

* "The similar convergence rate guarantees that PPD-VQA promises an intuitive linear runtime speedup concerning the increased number of local nodes " -> you mean the speedup of the computation of the gradient, right? Please be very clear here.

* "Even if the average noise of each quantum processor is the same,
the noise environment of the qubits executing quantum circuits in different processors is unlikely to be consistent" -> What do you mean by this statement? The hardware noise? But, then, why is this different from bullet point (1) you mentioned before this sentence? Also, please always provide references to such important messages.

In all, given what I have observed, I feel the paper is not mature enough for SciPost Physics, considering the acceptance criterion of the journal:

- Detail a groundbreaking theoretical/experimental/computational discovery;

- Present a breakthrough on a previously-identified and long-standing research stumbling block;

- Open a new pathway in an existing or a new research direction, with clear potential for multipronged follow-up work;

- Provide a novel and synergetic link between different research areas.

Requested changes

1. The author should consider how to answer my major questions 1 - 5.

2. Please go through the manuscript very carefully against grammatical issues, and improve the clarity of the writing.

---

## Editorial Decision

resubmitted